# MuG: A Multimodal Classification Benchmark on Game Data with Tabular, Textual, and Visual Fields

**Jiaying Lu*** and **Yongchen Qian*** and **Shifan Zhao** and **Yuanzhe Xi** and **Carl Yang**

Emory University

{jiaying.lu, yongchen.qian, shifan.zhao, yuanzhe.xi, j.carlyang}@emory.edu

## Abstract

Previous research has demonstrated the advantages of integrating data from multiple sources over traditional unimodal data, leading to the emergence of numerous novel multimodal applications. We propose a multimodal classification benchmark MuG with eight datasets that allows researchers to evaluate and improve their models. These datasets are collected from four various genres of games that cover tabular, textual, and visual modalities. We conduct multi-aspect data analysis to provide insights into the benchmark, including label balance ratios, percentages of missing features, distributions of data within each modality, and the correlations between labels and input modalities. We further present experimental results obtained by several state-of-the-art unimodal classifiers and multimodal classifiers, which demonstrate the challenging and multimodal-dependent properties of the benchmark. MuG is released at `https://github.com/lujiaying/MUG-Bench` with the data, tutorials, and implemented baselines.

## 1 Introduction

The world surrounding us is multimodal. Real-world data is often stored in well-structured databases that contain tabular fields, with textual and visual fields co-occurring. Numerous automated classification systems have been deployed on these multimodal data to provide efficient and scalable services. For instance, medical decision support systems (Soenksen et al., 2022) utilize patients' electronic health record data that contains tabular inputs (*e.g.*, ages, genders, races), textual inputs (*e.g.*, notes, prescriptions, written reports), and visual inputs (*e.g.*, x-rays, magnetic resonance imaging, ct-scans) to help precise disease prediction. Similarly, e-commerce product classification systems (Erickson et al., 2022) categorize products based on their categorical/numerical quanti-

---
*These authors contributed equally to this work

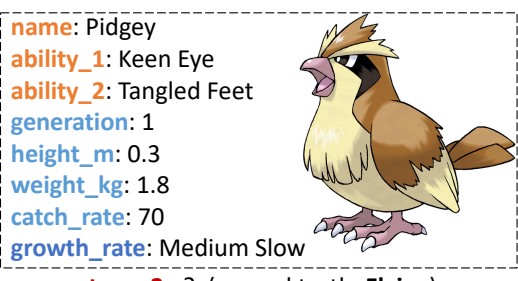

**type_2** =? (ground truth: **Flying**)

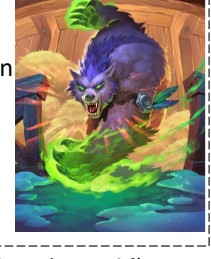

**cardClass** =? (ground truth: **Druid**)

Figure 1: Illustration of data examples from MuG. Note that inputs cover tabular, textual and visual modalities, and the task is multiclass classification.

ties, textual descriptions, and teasing pictures, thus enhancing user search experiences and recommendation outcomes. Therefore, accurate classification models for table-text-image input are desired.

Deep neural networks have shown significant progress in multimodal learning tasks, such as CLIP (Radford et al., 2021) for image-text retrieval and Fuse-Transformer (Shi et al., 2021) for tabular-with-text classification. This progress has been made with large-scale datasets provided to train the data-eager models. So far, there exist many datasets (Ovalle et al., 2017; Wu et al., 2021; Shi et al., 2021; Qu et al., 2022; Lin et al., 2020; Lee et al., 2019; Kautzky et al., 2020; Srinivasan et al., 2021) that cover one or two modalities. However, the progress in tabular-text-image multimodal learning lags due to the lack of available resources. In this paper, we provide a multimodal benchmark, namely MuG, that contains eight datasets for researchers to examine their al-

gorithms' multimodal perception ability. MuG contains data samples with tabular, textual, and visual fields that are collected from various genres of games. We have made necessary cleaning, transformations, and modifications to the original data to make MuG easy to use. We further conduct comprehensive data analysis to demonstrate the diverse and multimodal-dependent properties of MuG.

MuG can enable future studies of many multimodal tasks, and we focus on the multimodal classification task in this paper. For the primary classification evaluation, we incorporate two state-of-the-art (SOTA) unimodal classifiers for each of the three input modalities, resulting in a total of six, along with two SOTA multimodal classifiers. We also propose a novel baseline model MuGNET based on the graph attention network (Veličković et al., 2018). In addition to capturing the interactions among the three input modalities, our MuGNET takes the sample-wise similarity into account, yielding a compatible performance to existing multimodal classifiers. We further conduct efficiency evaluations to reflect the practical requirements of many machine learning systems.

## 2 Related Works

### 2.1 Multimodal Classification Datasets with Tabular, Textual, and Visual Fields

Machine learning models in real-world applications need to deal with multimodal data that contains both tabular, textual, and visual fields. Due to privacy or license issues, there exist very few datasets that cover these three modalities. To the best of our knowledge, PetFinder[1] is one of the few publicly available datasets. HAIM-MIMIC-MM (Soenksen et al., 2022) is a multimodal healthcare dataset containing tabular, textual, image, and time-series fields. However, only credentialed users can access HAIM-MIMIC-MM. On the other hand, there exist many datasets that cover two modalities (out of table, text, and image modalities). The most common datasets are the ones with both textual and visual features (MM-IMDB (Ovalle et al., 2017), V-SNLI (Vu et al., 2018), MultiOFF (Suryawanshi et al., 2020), WIT (Srinivasan et al., 2021), MELINDA (Wu et al., 2021), etc.). Meanwhile, (Shi et al., 2021; Xu et al., 2019; Qu et al., 2022) provide benchmark datasets for table and text modalities, For the combination of table and image

modalities, there are a bunch of datasets from the medical domain (Lin et al., 2020; Lee et al., 2019; Kautzky et al., 2020; Gupta et al., 2020). Other than the mentioned table, text, and image modalities, multimodal learning has also been conducted in time-series, speech, and video modalities (Zhang et al., 2022, 2020; Li et al., 2020).

### 2.2 Multimodal Classifiers for Tabular, Textual, and Visual Fields

Fusion is the core technology for multimodal classification problems, which integrates data from each input modality and utilizes fused representations for downstream tasks (classification, regression, retrieval, etc.). Based on the stage of fusion (Iv et al., 2021), existing methods can be divided into early, late, or hybrid fusion. Early fusion models (Sun et al., 2019; Shi et al., 2021; Zhu et al., 2021) usually fuse raw data or extracted features before they are fed into the learnable classifier, while late fusion models (Erickson et al., 2020; Soenksen et al., 2022; Lu et al., 2022) employ separate learnable encoders for all input modalities and then fuse these learned representations into the learnable classifier. Hybrid fusion models are more flexible, allowing for modality fusion to occur at different stages simultaneously (Qingyun et al., 2021; Li et al., 2021). Although existing works have demonstrated remarkable capability in modeling feature interactions, they ignore signals of sample proximity, such as the tendency for within a group to exhibit similar behavior or share common interests. In response, we propose our approach, MuGNET, which dynamically constructs graphs based on sample similarity and effectively combines graphical representation learning with multimodal fusion. Our approach draws inspiration from pioneering graph neural networks (Guo et al., 2021; Wang et al., 2021; Georgantas and Richiardi, 2022), which have achieved success in various classification tasks.

## 3 MuG: the benchmark

We create and release MuG with eight datasets for multimodal classification with tabular, text, and image fields to the community for future studies. Raw data and examples of how to appropriately load the data are provided in `https://github.com/lujiaying/MUG-Bench`. MuG is under the

---

[1] PetFinder: `https://www.kaggle.com/competitions/petfinder-adoption-prediction/overview`

"CC BY-NC-SA 4.0" license[2], and is designated to use for research purposes.

## 3.1 Data Sources

To collect multiple and large-scale datasets that support multimodal automated machine learning, we collected data from four games: Pokémon, Hearthstone, League of Legends, and Counter-Strike: Global Offensive. We deliberately chose these video games as they have distinct video game genres (*e.g.*, role-playing, card, multiplayer online battle arena, and shooting). All of these datasets were gathered from publicly accessible web content by October 2022, and there are no licensing issues associated with them. They do not contain any user-specific private information. In particular,

- **Pokémon** is a video game centered around fictional creatures called "Pocket Monsters" that trainers capture and train to battle each other. Pokémon is owned by *Nintendo Co., Ltd.*, *Creatures Inc.*, and *Game Freak Inc.* Pokémon data is collected from `https://bulbapedia.bulbagarden.net/wiki` under the "CC BY-NC-SA 2.5" license.
- **HearthStone** is an online collectible card game developed by *Blizzard Entertainment, Inc.*, featuring strategic gameplay where players build decks and compete against each other using a variety of spells, minions, and abilities. Hearthstone data is collected from `https://hearthstonejson.com/` under the "CC0" license.
- **League of Legends** (LoL) is a multiplayer online battle arena (MOBA) video game developed by *Riot Games, Inc.* where teams of players compete in fast-paced matches, utilizing unique champions with distinct abilities to achieve victory. LoL data is collected from `https://lolskinshop.com/product-category/lol-skins/`.
- **Counter-Strike: Global Offensive** (CS:GO) is a multiplayer first-person shooter video game developed by *Valve Corporation* and *Hidden Path Entertainment, Inc.*, where players join teams to compete in objective-based matches involving tactical gameplay and precise shooting. CS:GO data is collected from `https://www.csgodatabase.com/`.

---

[2]CC BY-NC-SA 4.0: `https://creativecommons.org/licenses/by-nc-sa/4.0/`

## 3.2 Creation Process

To create MUG, we first identify the categorical columns that can serve as the prediction targets. The reasons for choosing these targets are elaborated in Appendix B.1. We obtain a total of eight datasets from the four games, including **pkm_t1** and **pkm_t2** from Pokémon; **hs_ac**, **hs_as**, **hs_mr**, and **hs_ss**; **lol_sc** from LoL; **csg_sq** from CS:GO.

Then, we conduct necessary data cleaning and verification to ensure the quality of MUG. To alleviate the class imbalance issue in some datasets (*e.g.*, one class may contain less than 10 samples), we re-group sparse classes into one new class that contains enough samples for training and evaluation. For missing values of target categorical columns, we manually assign a special *None_Type* as one new class of the dataset. For missing values of input columns, we keep them blank to allow classification models to decide the best imputation strategies. Moreover, we also anonymize columns that cause data leakage (*e.g.*, the *id* column in hs_as is transformed to *anonymous_id* column).

After the abovementioned preprocessing, we split the dataset into training, validation, and testing sets with an 80/5/15 ratio. Each dataset compromises between 1K and 10K samples, associated with tabular, textual, and visual features. An overview of these datasets is shown in Table 1. This broad range of sample sizes and diverse data types ensures the representation of a wide variety of instances, allowing for robust model training and evaluation across different data modalities.

## 3.3 Benchmark Analysis

The MUG benchmark is curated to meet the following list of criteria:

(i) **Publicly available** data and baseline models can facilitate reproducible experiments and accelerate the development of advanced models.

(ii) **Diversity** should be preserved in the benchmark. We do not want the benchmark to have a bias toward certain data or class distribution. The benchmark with a high variety of datasets aids the research community in examining the robustness of models.

(iii) **Multimodal-dependent** classification is expected for each dataset. Datasets that are too easy to be classified by a single modality are not suitable, since they would hide the gap between the multimodal perceptron abilities of models.

We conduct a rich set of analyses to verify

| Dataset | Game | Pred. Target | #Row | #Class | #Feat (tab/txt/img) |
|---------|------|--------------|------|--------|---------------------|
| pkm_t1 | Pokémon | Primary **T**ype | 719/45/133 | 18 | 23 (17/5/1) |
| pkm_t2 | Pokémon | Secondary **T**ype | 719/45/133 | 19 | 23 (17/5/1) |
| hs_ac | HearthStone | **A**ll card's **C**ategory | 8569/536/1605 | 14 | 18 (12/5/1) |
| hs_as | HearthStone | **A**ll card's **S**et | 8566/533/1607 | 38 | 18 (12/5/1) |
| hs_mr | HearthStone | **M**inion card's **R**ace | 5421/338/1017 | 16 | 13 (7/5/1) |
| hs_ss | HearthStone | **S**pell card's **S**chool | 2175/170/508 | 8 | 11 (5/5/1) |
| lol_sc | LoL | **S**kin **C**ategory | 1000/64/188 | 7 | 11 (3/7/1) |
| csg_sq | CS:GO | **S**kin **Q**uality | 766/49/141 | 6 | 7 (5/1/1) |

Table 1: The statistics of the eight datasets in MᴜG.

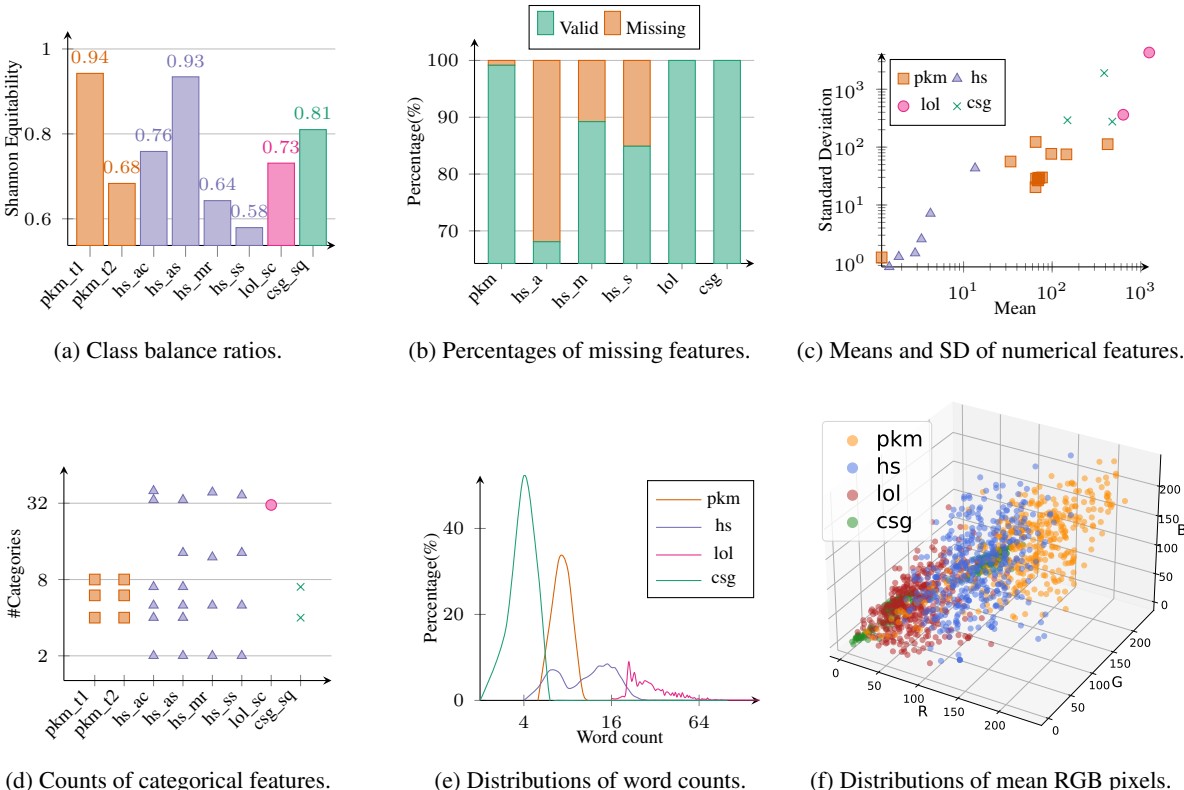

(a) Class balance ratios.  (b) Percentages of missing features.  (c) Means and SD of numerical features.

(d) Counts of categorical features.  (e) Distributions of word counts.  (f) Distributions of mean RGB pixels.

Figure 2: Multi-aspect of data analysis for MᴜG (duplicated datasets are merged into one group).

that MᴜG indeed satisfied the diversity requirement. Figure 2 shows the properties of datasets in multi-aspect. For the classification task properties(Figure 2a), we adopt the Shannon equitability index (Shannon, 1948) (definition in Appendix B.2) to measure the class balance ratio. The index ranges from 0 to 1, and the larger the Shannon equitability index, the more balanced the dataset is. For the feature properties, we include percentages of missing features (Figure 2b), means and standard deviations of numerical features (Figure 2c), category counts of categorical features (Figure 2d), distributions of word counts per sample (Figure 2e),

and distributions of image mean RGB pixel values (Figure 2f). In these figures, we merged duplicated results from some datasets into one group to make the presentation clean and neat (*i.e.*, *pkm_t1, pkm_t2* are grouped into *pkm*; *hs_ac, hs_as, hs_mr, hs_ss* are grouped into *hs*). As shown in the figures, the eight datasets reflect real-world problems that are diverse and challenging. We further study the correlation between category labels and input modalities in MᴜG. Referring to the t-SNE projection of multimodal embeddings in Figure 7, it is evident that MUG exhibits a strong multimodal dependency. In this case, the use of unimodal in-

formation alone is inadequate to differentiate between samples belonging to different classes. For a more comprehensive analysis, we encourage interested readers to refer to the details provided in Appendix B.3

## 4 Baseline Models

We employ several state-of-the-art unimodal classifiers and multimodal classifiers in the experiments. We also proposed our own graph neural network-based multimodal classifier as one baseline model to be compared.

### 4.1 Existing State-Of-The-Art Classifiers

In this paper, we adopt the following SOTA *unimodal classifiers* in the experiments:

**Tabular modality classifiers**:

- **GBM** (Ke et al., 2017) is a light gradient boosting framework based on decision trees. Due to its ability to capture nonlinear relationships, handle complex tabular data, provide feature importance insights, and robustness to outliers and missing values, GBM has achieved state-of-the-art results in various tabular data tasks,
- **tabMLP** (Erickson et al., 2020) is a multilayer perceptron (MLP) model that is specifically designed to work with tabular data. tabMLP contains multiple separate embedding layers to handle categorical and numerical input features.

**Textual modality classifiers**:

- **RoBERTa** (Liu et al., 2019) is a robustly optimized transformer-based masked language model (masked LM). RoBERTa builds upon the success of BERT by refining and optimizing its training methodology, and achieves superior performance on a wide range of NLP tasks.
- **Electra** (Clark et al., 2020) is another variant of the transformer-based model, which differs from traditional masked LMs like BERT or RoBERTa. While masked LMs randomly mask tokens and predict these masked tokens, Electra is trained as a discriminator to identify whether each token is replaced by a generator.

**Visual modality classifiers**:

- **ViT** (Dosovitskiy et al., 2020) extends the transformer model to image data, by dividing the input image into a grid of patches and processing each patch as a token. Empirical results show that ViT outperforms previous SOTA convolutional neural networks in image classification tasks.

- **SWIN** (Liu et al., 2021) is another vision transformer that benefits from hierarchical architecture and the shifted windowing scheme. The proposed techniques address several key challenges when adapting transformers in image modality, such as large variations in the scale of visual entities and the high resolution of pixels.

In practice, we adopt the following *multimodal classifiers* in the experiments:

- **AutoGluon** (Erickson et al., 2022) is an ensemble-learning model for multimodal classification and regression tasks. The concept of AutoGluon is stack ensembling, where the final prediction is obtained by combining intermediate predictions from multiple base models. To handle multimodal classification, SOTA unimodal classifiers (*e.g.*, tree models, MLPs, CNNs, transformers) are adopted as base models.
- **AutoMM** (Shi et al., 2021) is a late-fusion model where separate neural operations are conducted on each data type and extracted high-level representations are aggregated near the output layer. Specifically, MLPs are used for tabular modality, and transformers are used for text and image modalities. After that, dense vector embeddings from the last layer of each network are pooled into one vector, and the final prediction is obtained via an additional two-layer MLP.

### 4.2 MUGNET

MUGNET is our own multimodal classifier which is further proposed as a competitor to existing models. We propose three key components to make MUGNET a powerful graph neural network for the multimodal classification task. They are adaptive multiplex graph construction module, GAT encoder module, and attention-based fusion module, as shown in Figure 3. Firstly, adaptive multiplex graphs are constructed to reflect sample-wise similarity within each modality. Then, separate GAT encoders (Veličković et al., 2018) are employed to obtain dense embeddings of samples, by propagating information between neighbors. Finally, tabular, text and image embeddings are combined by inter-modality attention to obtaining the fused embedding for multimodal classification. GNNs (Yang et al., 2020; Guo et al., 2021) show great capability to leverage the graph structure, propagate information, integrate features, and capture higher-order relationships. This leads to accurate and robust classification performance across various domains.

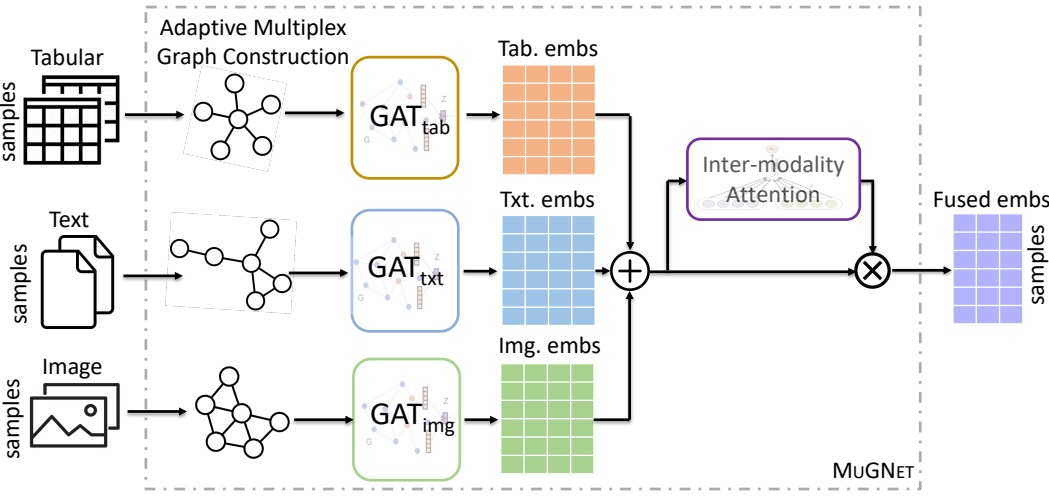

Figure 3: Model architecture of MUGNET.

In this work, we propose to regard the whole samples as a correlation network (Wang et al., 2021; Georgantas and Richiardi, 2022) that represents sample-to-sample similarities, while existing multimodal classifiers rarely consider this before.

**Adaptive multiplex graph construction module**. Following the notation defined in §5.1, the adaptive multiplex graph construction module first utilizes pre-processing pipelines (*e.g.*, monotonically increasing integer mapping for categorical inputs, no alteration for numerical inputs) or pre-trained feature extractors (*e.g.*, CLIP (Radford et al., 2021) for text and image inputs) to obtain dense multimodal features $\mathcal{F} = f(\mathcal{X}_L) \in \mathbb{R}^{N \times (d^t + d^s + d^i)}$, where $\mathcal{F} = \{\mathcal{F}^t, \mathcal{F}^s, \mathcal{F}^i\}$ denotes feature matrices for tabular, text, and image modalities. The adaptive multiplex graph construction module then derives multiplex sample-wise similarity graph $\mathcal{G} = \{\mathcal{G}^t, \mathcal{G}^s, \mathcal{G}^i\} = \{(\mathcal{A}^t, \mathcal{F}^t),$ $(\mathcal{A}^s, \mathcal{F}^s), (\mathcal{A}^i, \mathcal{F}^i)\}$, where each modality-specific adjacency matrix $\mathcal{A}^m \in \mathbb{R}^{N \times N}, \forall m \in \{t, s, i\}$ is calculated based on the multimodal features

$$\mathcal{A}^m_{i,j} = \text{sim}(\mathcal{F}^m_i, \mathcal{F}^m_j). \tag{1}$$

It is worth noting that the sample-wise similarity function $\text{sim}$ is adaptive, and is chosen from *cosine similarity*, *radial basis function (RBF) kernel*, or *k-nearest neighbor*. For these modality-specific graphs, we use separate hyperparameters (*e.g.*, threshold for score-based functions, or the value of $k$ for k-nearest neighbor) to control their sparsity properties. The similarity function and its associated hyperparameters are determined through hyperparameter tuning (Liaw et al., 2018) on the

held-out validation set, so that the multiplex graph construction is adaptive to any downstream task.

**GAT encoder module.** We use the powerful multi-head graph attention neural network (GAT) (Veličković et al., 2018) as the encoder to obtain structure-aware representations of samples. Separate GATs are employed for each view of the multiple graph, so that $\mathcal{H}^m = GAT(\mathcal{A}^m, \mathcal{F}^m; \theta)$, where $\mathcal{H}^m \in \mathbb{R}^{N \times d^m_h}$, and $\theta$ represents the learnable parameters of the GAT encoder. We want to state there is no information leakage in MUGNET, because we follow the inductive learning setting of GNNs (Hamilton et al., 2017) where the GAT encoder is trained on the multiplex graph $\mathcal{G}$ derived from labeled training samples $\mathcal{X}_L$, and new unseen multiplex graph is derived from all samples $\mathcal{X}_L \cup \mathcal{X}_U$ at the inference stage. Furthermore, we adopt a graph sampling technique (Graph-SAINT (Zeng et al., 2019)) during the GAT training process, to improve the efficiency and generalization. The graph sampling technique essentially samples a subgraph by random walks for each training step, thus the "neighbor explosion" issue is alleviated with a constrained number of neighbors per node and the variance of GAT is reduced with fewer outliers or noise in the sampled graph.

**Attention-based fusion module.** After we obtain the structure-aware embeddings of samples from the tabular, text, and image modalities $\mathcal{H}^t, \mathcal{H}^s, \mathcal{H}^i$, the attention-based fusion module is responsible for fusing them into one single embedding via the attention-based fusion module. The attention weight $\alpha^m_j \in \mathbb{R}$ for $j$-th sample of modality $m$ is

computed as:

$$\alpha_j^m = \frac{\exp(e_j^m)}{\sum_{m' \in \{t,s,j\}} \exp(e_j^{m'})}, \qquad (2)$$

$$e_j^m = \boldsymbol{w}_{a_2} \cdot \tanh(\boldsymbol{W}_{a_1}^m \boldsymbol{h}_j^m), \qquad (3)$$

where $e_j^m \in \mathbb{R}$ denotes the unnormalized attention weight, $\boldsymbol{w}_{a_2} \in \mathbb{R}^{d_a^m \times 1}$, $\boldsymbol{W}_{a_1} \in \mathbb{R}^{d_h^m \times d_a^m}$ denote learnable parameters, and $\boldsymbol{h}_j^m \in \mathbb{R}^{d_h^m}$ denotes the $j$-th row of $\mathcal{H}^m$ (*i.e.*, embedding of $j$-th sample of modality $m$). The fused embedding of $j$-th sample is then calculated by:

$$\boldsymbol{h}_j = \alpha_j^t \boldsymbol{h}_j^t + \alpha_j^s \boldsymbol{h}_j^s + \alpha_j^i \boldsymbol{h}_i^i. \qquad (4)$$

The fused embedding $\boldsymbol{h}_j$ incorporates cross-modalities interactions and provides a complete context for the downstream tasks. An additional two-layer MLP is trained to predict the category of $j$-th sample $\hat{y}_j = \mathrm{softmax}(\boldsymbol{W}_{\mathrm{cls}_2} \cdot \mathrm{LeakyReLU}(\boldsymbol{W}_{\mathrm{cls}_1} \boldsymbol{h}_j))$. We adopt cross-entropy between prediction $\hat{y}$ and target $y$ as MUGNET's loss function.

## 5 Experiments

### 5.1 Problem Definition

Given a finite set of categories $\mathcal{Y}$ and labeled training pairs $(x_i, y_i) \in \mathcal{X}_L \times \mathcal{Y}$, multimodal classification aims at finding a classifier $\hat{f} : \mathcal{X}_L \to \mathcal{Y}$ such that $\hat{y}_j = \hat{f}(x_j)$ is a good approxmiation of the unknown label $y_j$ for unseen sample $x_j \in \mathcal{X}_U$. It is worth noting that the each multimodal sample $x \in \mathcal{X}_L \cup \mathcal{X}_U$ consists of tabular fields $t$, textual fields $s$, and image fields $i$ (*i.e.*, $x = \{t, s, i\}$).

### 5.2 Experimental Setup

We use the official training, validation, and testing splits provided by MUG to conduct experiments. We choose the log-loss and accuracy to evaluate model performance, since these metrics are reasonable and commonly used in previous studies. For comparable and reproducible results, all models are trained and tested using the same hardware. Specifically, the machine is equipped with 16 Intel Xeon Gold 6254 CPUs (18 cores per CPU) and one 24GB TITAN RTX GPU. We add an 8-hour time limitation for the training process to reflect real-world resource constraints. The implementation and hyperparameter details of evaluated models are put in Appendix C.

### 5.3 Performance Comparisons

Table 2a and 2b show the performance of all evaluated models on MUG. As can be seen, multimodal classifiers (except AutoMM) consistently outperform unimodal classifiers in both log-loss and accuracy. It demonstrates that the classification tasks in MUG are multimodal-dependent where each modality only conveys partial information about the required outputs. Among the three multimodal classifiers we used, AutoGluon and MUGNET are the top-2 models with well-matched performances. In Table 2a and 2b, AutoGluon achieves the best performance eight times, while MUGNET also achieves the best performance eight times. More specifically, AutoGluon is superior in log-loss whereas MUGNET has better accuracy scores. AutoMM performs the worst among multimodal classifiers, and it sometimes underperforms unimodal classifiers. Considering that AutoMM trains powerful deep neural networks on a small scale of datasets and we have observed the gap between the training loss and validation loss, it is highly possible that AutoMM is overfitting. While AutoGluon and MUGNET also adopt deep neural networks as base models, they are more robust since AutoGluon proposes a repeated bagging strategy and MUGNET utilizes graph sampling techniques to avoid overfitting. Among unimodal classifiers, tabular models seem to outperform textual and visual models in most cases (six out of eight datasets). There is a slight performance gain comparing textual models to visual models because textual models are better on five datasets.

To better understand the overall performance of models across multiple datasets, we propose using critical difference (CD) diagrams (Demšar, 2006). In a CD diagram, the average rank of each model and which ranks are statistically significantly different from each other are shown. Figure 4a and 4b show the CD diagrams using the Friedman test with Nemenyi post-hoc test at $p < 0.05$. In summary, we observe that *AutoGluon* and MUGNET respectively achieve the best rank among all tested models with respect to log-loss and accuracy, although never by a statistically significant margin. Moreover, tabular models obtain higher ranks than other unimodal classifiers. The similar observations from Table 2 and Figure 4 support that effectively aggregating information across modalities is critical for the multimodal classification task.

| Method | pkm_t1 | pkm_t2 | hs_ac | hs_as | hs_mr | hs_ss | lol_sc | csg_sq |
|---|---|---|---|---|---|---|---|---|
| | | | Unimodal Classifiers | | | | | |
| GBM | 1.838 | 2.038 | 0.911 | 2.352 | 0.913 | 0.603 | 0.198 | 1.107 |
| tabMLP | 1.442 | 1.909 | 1.172 | 2.155 | 1.247 | 0.672 | 0.533 | 0.718 |
| RoBERTa | 1.834 | 2.191 | 1.999 | 2.393 | 1.920 | 1.254 | 0.847 | 0.734 |
| Electra | 2.907 | 2.179 | 2.118 | 3.155 | 2.085 | 1.263 | 0.611 | 0.757 |
| ViT | 3.680 | 2.543 | 1.527 | 2.786 | 1.032 | 2.056 | 2.049 | 0.835 |
| Swin | 2.657 | 2.229 | 2.018 | 2.795 | 2.089 | 1.397 | 1.470 | 0.750 |
| | | | Multimodal Classifiers | | | | | |
| AutoGluon | **0.973** | 1.507 | **0.654** | 1.793 | 0.403 | **0.350** | **0.159** | **0.631** |
| AutoMM | 1.736 | 2.029 | 1.987 | 2.193 | 1.836 | 1.320 | 0.792 | 0.674 |
| MᴜGNᴇᴛ | 1.000 | **1.499** | 0.922 | **1.499** | **0.321** | 0.442 | 0.248 | 0.654 |

(a) Results in 'log-loss' (the less the better).

| Method | pkm_t1 | pkm_t2 | hs_ac | hs_as | hs_mr | hs_ss | lol_sc | csg_sq |
|---|---|---|---|---|---|---|---|---|
| | | | Unimodal Classifiers | | | | | |
| GBM | 0.489 | 0.489 | 0.726 | 0.421 | 0.737 | 0.795 | 0.963 | 0.610 |
| tabMLP | 0.662 | 0.481 | 0.627 | 0.377 | 0.617 | 0.776 | 0.851 | 0.681 |
| Roberta | 0.662 | 0.466 | 0.475 | 0.366 | 0.535 | 0.683 | 0.883 | 0.688 |
| Electra | 0.120 | 0.466 | 0.475 | 0.168 | 0.535 | 0.683 | 0.878 | 0.702 |
| ViT | 0.308 | 0.406 | 0.568 | 0.236 | 0.787 | 0.593 | 0.436 | 0.674 |
| Swin | 0.346 | 0.451 | 0.470 | 0.248 | 0.536 | 0.657 | 0.431 | 0.702 |
| | | | Multimodal Classifiers | | | | | |
| AutoGluon | 0.744 | 0.617 | **0.787** | 0.495 | 0.879 | **0.882** | 0.963 | **0.766** |
| AutoMM | 0.639 | 0.511 | 0.475 | 0.415 | 0.549 | 0.671 | 0.888 | 0.738 |
| MᴜGNᴇᴛ | **0.774** | **0.669** | 0.724 | **0.572** | **0.908** | 0.880 | **0.968** | 0.745 |

(b) Result in 'accuracy' (the more the better).

Table 2: Overall experimental results with explicit modality performance. The bold text represents the best performance and the underlined text represents the runner-up performance.

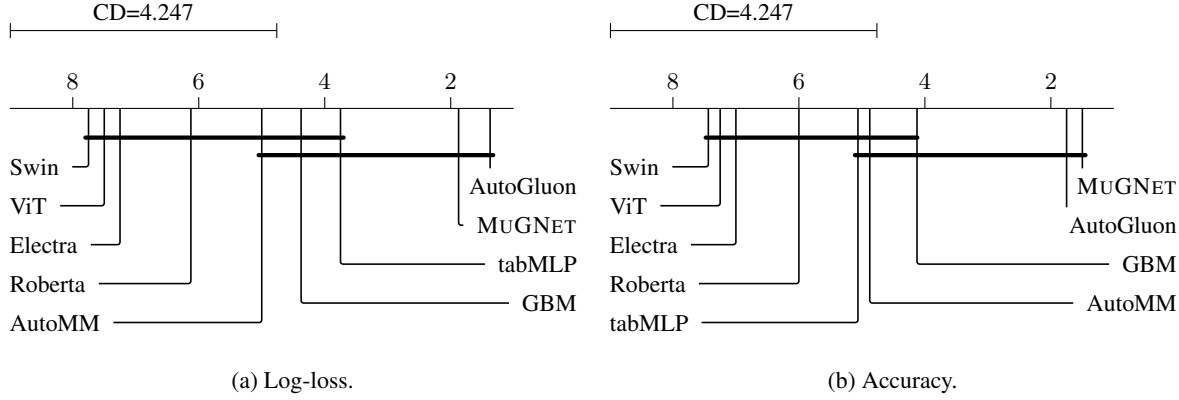

(a) Log-loss.

(b) Accuracy.

Figure 4: The critical difference diagrams show the mean ranks of each model for the test data of the eight datasets. The lower rank (further to the right) represents the better performance of a model. Groups of models that are not significantly different ($p < 0.05$) are connected by thick lines.

## 5.4 Efficiency Evaluations

Although accuracy (or other metrics such as log-loss in our case) is the central measurement of a machine learning model, efficiency is also a practical requirement in many applications. Trade-off often exists between *how accurate* the model is

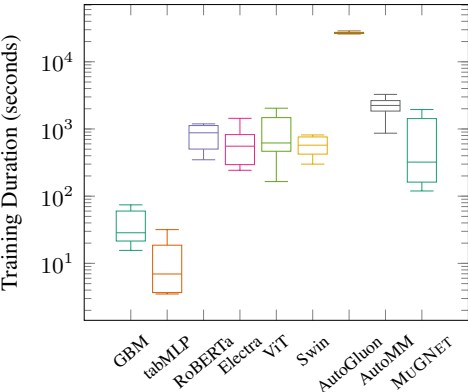

Figure 5: Training duration on all datasets.

and *how long* it takes to train and infer the model. Therefore, we record the training durations and test durations of models to examine their efficiency. In Figure 5, we show the aggregated training duration of evaluated models via a box plot. As can be seen, tabular models require an order of magnitude less training duration than the other models, while AutoGluon stands out as requiring significantly longer training duration. Among tabular models, tabMLP is 4x faster than GBM in terms of the median training duration. Except for tabular models and AutoGluon, other models are approximately lightweight to train. It is worth noting that AutoGluon hits the 8-hour training duration constraint on every dataset, thus the variance of its training durations across datasets is very small.

In Figure 6, we show the trade-offs between mean inference time and mean accuracy of models. Since the accuracy is not commensurable across datasets, we first normalize all accuracies through a dataset-wise min-max normalization. After the normalization, the best model in each dataset is scaled to 1 while the worst model is scaled to 0. Finally, we take the average on the normalized accuracies and the test durations to draw the scatter plot. When both accuracy and efficiency are objectives models try to improve, there does not exist a model that achieves the best in both objectives simultaneously. As an illustration, MUGNET has the highest test accuracy, but tabMLP has the fastest inference speed. Therefore, we adopt the Pareto-optimal[3] concept to identify which models achieve "optimal" trade-offs. Pareto-optimal is widely used in the decision-making process for multi-objective optimization scenarios. By definition, a solution is Pareto-optimal if any of the objectives cannot

---

be improved without degrading at least one of the other objectives. Following this concept, we observe that tabMLP, GBM, and MUGNET are the models with the best trade-offs between accuracy and efficiency, as these models reside in the Pareto frontier in Figure 6. Meanwhile, other models are suboptimal with regard to this trade-off, since we can always find a solution that has higher accuracy and better efficiency simultaneously than these models.

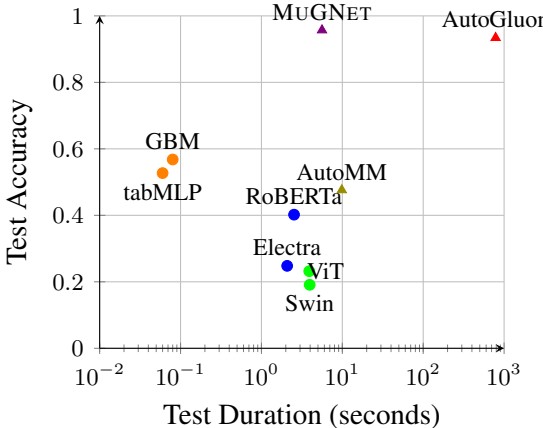

Figure 6: Mean testing duration and mean normalized accuracy tradeoffs on all datasets.

## 6 Conclusion

This paper presents a benchmark dataset MUGNET along with multiple baselines as a starting point for the machine learning community to improve upon. MUGNET is a multimodal classification benchmark on game data that covers tabular, textual, and visual modalities. All eight datasets and nine evaluated baselines are open-source and easily-extended to motivate rapid iteration and reproducible experiments for researchers. A comprehensive set of analyses is included to provide insight into the characteristics of the benchmark. The experimental results reported in the paper are obtained from models trained with constrained resources, which is often required by real-life applications. However, we also welcome future works that utilize enormous resources. Finally, we hope this work can facilitate research on multimodal learning, and we encourage any extensions to MUGNET to support new tasks or applications such as open-domain retrieval, AI-generated content, multimodal QA, etc.

---

[3]Pareto-optimal Definition: `https://w.wiki/6sLB`

## Limitations

While our study emphasizes the importance of efficiency in real-world machine learning applications, we acknowledge certain limitations in our approach. Specifically, we deliberately focused on training and evaluating relatively "small" models within the context of the current era of large vision and language models (LVLMs) (Li et al., 2023; Liu et al., 2023; Lu et al., 2023). As a result, the performance of LVLMs on our proposed MUG benchmark remains unexplored. Early exploration (Hegselmann et al., 2023) about applying large language models on tabular classification shows that LLMs can be competitive with strong tree-based models. Based on the explorations and conducted experiments, we speculate LLVMs can not beat efficient ensemble or GNN baselines using the same training time constraint. However, to provide a comprehensive understanding of multimodal classification, further research is expected. It would be also intriguing to investigate the performance of LLVMs when provided with unlimited training (fine-tuning) time.

## Acknowledgement

This research is partly supported by the National Institute Of Diabetes And Digestive And Kidney Diseases of the National Institutes of Health under Award Number K25DK135913 and the Division of Mathematical Sciences of the National Science Foundation under Award Number 2208412. Any opinions, findings, and conclusions or recommendations expressed herein are those of the authors and do not necessarily represent the views, either expressed or implied, of the National Science Foundation, National Institutes of Health, or the U.S. government.

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

# Appendix

## A  Broad Impact of Multimodal Datasets for Tasks beyond Classification

The proposed multimodal classification benchmark, MUG, is a valuable resource to inspire future studies for tasks including but not limited to classification. First, it is natural to utilize MUG to examine the model's abilities to understand the complex world surrounding us (Liang et al., 2021; Xu et al., 2022). The multimodal perception ability is crucial for open-domain retrieval (Tahmasebzadeh et al., 2021; Tian et al., 2022; Zeng et al., 2021), multimodal classification (Zadeh et al., 2017; Huddar et al., 2018), multimodal question answering (Antol et al., 2015; Talmor et al., 2020; Lu et al., 2022), interactive robots (Nie et al., 2021; Sampat et al., 2022), precision medicine (Soenksen et al., 2022; Lin et al., 2020; Gupta et al., 2020), *etc*. These applications all involve multi-modal input where each modality conveys partial information and a complete understanding can be achieved by taking all modalities into account. Second, MUG that contains aligned tabular, textual, and visual data is also beneficial to multimodal generation applications (Dong et al., 2023; He et al., 2023; Sun et al., 2023; Peng et al., 2023). For instance, many text-to-image generation models (Ramesh et al., 2021; Poole et al., 2022) employ CLIP (Radford et al., 2021) as their feature encoder, and CLIP is an alignment-based fusion model trained on semantically equivalent text and image pairs. There also exist many studies exploring unimodal-to-unimodal generation tasks, such as image-to-text captioning (Hossain et al., 2019), table-to-text

generation (Parikh et al., 2020), text-to-table generation (Wu et al., 2022), etc. Following this idea, the aligned triples can be utilized to pre-train multimodal encoders. Furthermore, multimodal datasets that cover more than two modalities can inspire more novel generation applications, such as audio-visual slideshows generation from text (Leake et al., 2020) or textual-visual summarization from video-based news articles (Li et al., 2020).

## B  Details of MUG

### B.1  Prediction Targets

| Dataset | Source | Prediction Target |
|---------|--------|-------------------|
| pkm_t1 | Pokémon | Pokémon's primary **T**ype |
| pkm_t2 | Pokémon | Pokémon's secondary **T**ype |
| hs_ac | HearthStone | **A**ll card's **C**ategory |
| hs_as | HearthStone | **A**ll card's **S**et |
| hs_mr | HearthStone | **M**inion card's **R**ace |
| hs_ss | HearthStone | **S**pell card's **S**chool |
| lol_sc | LoL | **S**kin **C**ategory |
| csg_sq | CS:GO | **S**kin **Q**uality |

Table 3: The prediction targets of datasets in MUG.

We identify appropriate categorical columns that can serve as the prediction targets from these four games, with a handy reference presented in Table 3. More specifically, we provide detailed elaborations about the prediction targets and their corresponding input multimodal features. Some MUG datasets may share same input multimodal features.

- **pkm_t1** and **pkm_t2**: Pokémon can be categorized into various elemental types, such as Fire, Ice, Normal (non-elemental), and more. Each Pokémon can have up to one primary type (for pkm_t1) and one secondary type (for pkm_t2).

  – 17 tabular features: *generation, status, height_m, weight_kg, abilities_num, total_points, hp, attack, defense, sp_attack, sp_defense, speed, catch_rate, base_friendship, base_experience, growth_rate, percentage_male.*
  – 5 text features: *name, species, ability_1, ability_2, ability_hidden.*
  – 1 image feature: *image.*

- **hs_ac** and **hs_as**: Each HearthStone card belongs to one certain category (for hs_ac) such as minion, spell, weapon, etc. Moreover, each card is part of a set (for hs_as) where new card sets are

released periodically to introduce new content and strategies to the game.

- 12 tabular features: *health, attack, cost, type, rarity, collectible, spellSchool, race, durability, overload, spellDamage, set (for hs_ac) / cardClass (for hs_as).*
- 5 text features: *name, id, artist, text, mechanics.*
- 1 image feature: *image.*

- **hs_mr**: Each HeathStone minion card is essentially a creature, thus it can be divided into different races (for hs_mr).

  - 7 tabular features: *health, attack, cost, rarity, collectible, cardClass, set.*
  - 5 text features: *name, id, artist, text, mechanics.*
  - 1 image feature: *image.*

- **hs_ss**: Each HeathStone spell card, similarly to the minion race, each spell card belongs to a specific school (for hs_ss) such as Shadow, Nature, etc.

  - 5 tabular features: *cost, rarity, collectible, cardClass, set, attack.*
  - 5 text features: *name, id, artist, text, mechanics.*
  - 1 image feature: *image.*

- **lol_sc**: A champion skin in LoL is a cosmetic alteration to the appearance of the champion. Depending on the rarity and price, a champion's skin belongs to a specific category (for lol_sc). It is worth noting that the champion skins are stylish decorations that have nothing to do with race, gender, or other unethical variables.

  - 3 tabular features: *id, price, soldInGame.*
  - 7 text features: *skinName, concept, model, particles, animations, sounds, releaseDate.*
  - 1 image feature: *image.*

- **csg_sq**: Similar to champion skin in LoL, CS:GO skins alter the appearance of weapons. The prediction target is the skin quality according to its rarity (for csg_sq).

  - 5 tabular features: *id, availability, skinCategory, minPrice, maxPrice.*
  - 1 text features: *skinName.*
  - 1 image feature: *image.*

## B.2 Definition of Shannon Equitability Index.

The Shannon equitability index (Shannon, 1948), also known as the Shannon evenness index or Shannon's diversity index, is a measure used in ecology to assess the evenness or equitability of species abundance in a given community or ecosystem. It is derived from the Shannon entropy, which quantifies the diversity or richness of species in a community. For the classification task properties (Figure 2a), we adopt it to measure the class balance ratio,

$$E_H = \frac{H}{\log(k)} = \frac{-\sum_{i=1}^{k} \frac{c_i}{n} \log \frac{c_i}{n}}{\log(k)}, \quad (5)$$

where $H$ denotes the entropy of each class's counts, $k$ denotes the dataset containing $k$ classes. $E_H$ ranges from $0$ to $1$, and the large $E_H$, the more balanced the dataset is.

## B.3 Analysis on Multimodal-dependent.

To study the correlation between category labels and input modalities in MuG, we plot 2D t-SNE (Van der Maaten and Hinton, 2008) projections of various embeddings for the *hs_mr* dataset. In the first row of Figure 7, the four subfigures present projections obtained from the raw features of tabular, textual, visual, and fused modalities, separately. Essentially, we conduct unsupervised dimension reduction (*e.g.*, SVD) on the raw features and then use t-SNE to obtain 2D projections. For tabular features, numerical columns are kept as they are, and categorical columns are transformed into numbers between 0 and $n\_class - 1$. For textual features, we first transformed them into token count vectors, and then use TruncatedSVD (Pedregosa et al., 2011) to reduce the number of dimensions to a reasonable amount (*e.g.*, 50) before feeding into t-SNE. For visual features, we conduct PCA (Pedregosa et al., 2011) on each color channel to reduce the number of dimensions (*e.g.*, 30) as well.

For a neat presentation, we select a subgroup of categories in the *hs_mr* dataset and assign different colors to samples belonging to different categories. For the fused raw features, we simply concatenate the three single modality features without any further modifications. The 2D projection of fused features is obtained following the same procedure, as in unimodal features. As can be seen in these four subfigures, samples from different categories are clustered together no matter what modality is used as the input. The second row of Figure 7 shows the

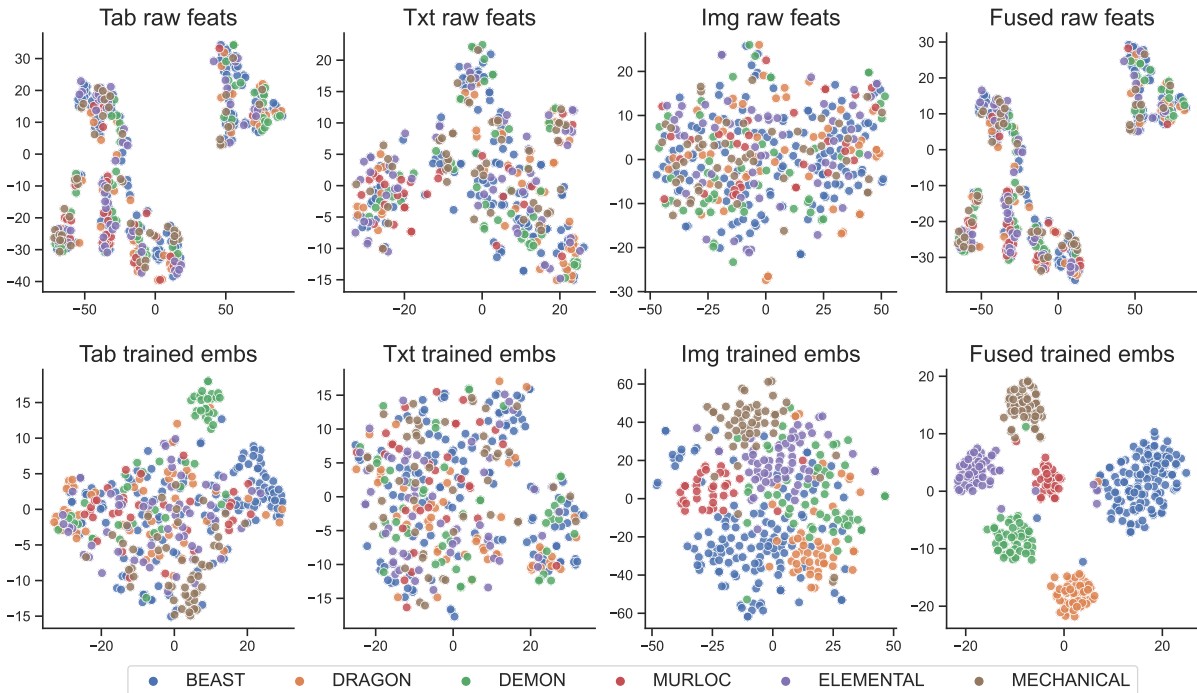

Figure 7: 2D t-SNE projections of raw features (the 1st row) and trained embeddings (the 2nd row), based on unimodal (1st to 3rd columns) or multimodal (the 4th column) inputs.

2D t-SNE projections obtained from embeddings of models trained on tabular, textual, visual, and fused modalities, where these embeddings are obtained from the output of the penultimate layers. The models we used are tabMLP, RoBERTa, Vit, and MUGNET (evaluated baselines as in Sec 4). Compared to the t-SNE projections using raw features, all projections using trained embeddings provide better insights into categorical structures of the data. Among all subfigures, the one using fused embeddings is significantly better than the others, in which the separation between different categories is almost perfect with only a small number of points mis-clustered. In summary, MUG is multimodal-dependent that requires multimodal information to distinguish samples from different classes.

## C  Implementation and Hyperparameters of Baselines

We implement the evaluated models using open-source codebases (Ke et al., 2017; Paszke et al., 2019; Wang, 2019; Erickson et al., 2020; Shi et al., 2021; Erickson et al., 2022), and models' hyperparameters without specification are set as default. For GBM, we set the maximum number of leaves in one tree as 128, the minimal number of data inside one bin as 3, and the feature fraction ratio in one tree as 0.9. For tabMLP, we follow (Erickson et al., 2020) to adaptively set the

embedding dimension of each categorical feature as $\min(100, 1.6 * \text{num\_cates}^{0.56})$, all hidden layer sizes as 128, and the number of layers as 4. For RoBERTa, we use the "RoBERTa-base" variant. For Electra, we use the "Electra-base-discriminator" variant. For ViT, we use the "vit_base_patch32_224" variant. For SWIN, we use the "swin_base_patch4_window7_224" variant. For AutoGluon, we use its "multimodal-best_quality" preset. For AutoMM, we use its default preset. For our own MUGNET, we optimize it using AdamW with a learning rate set as 0.001 and a cosine annealing learning rate schedule. Regarding the graph sampling strategy, we set the number of root nodes to generate random walks as 80% of the original number of nodes, and the length of each random walk as 2. MUGNET chooses other hyperparameters via HPO (Liaw et al., 2018). The search space includes the sample-wise similarity function $\text{sim} \in \{cosine\ sim, RBF\ kernel, k\text{-}nearest\ neighbor\}$ used in Equation (1). More specifically, (i) when $\text{sim} := cosine\ sim$ or $\text{sim} := RBF\ kernel$, HPO of MUGNET also search along their associated *graph sparsity* hyperparameter $spy \in \{0.5, 0.75, 0.95\}$. (ii) when $\text{sim} := k\text{-}nearest\ neighbor$, MUGNET search along $k \in \{5, 10, 32\}$.