# OpenReview forum: "MuG: A Multimodal Classification Benchmark on Game Data with Tabular, Textual, and Visual Fields"
_EMNLP/2023/Conference — EMNLP 2023 Findings_

### Official Review · Reviewer_L86v · 2023-08-05

**Soundness:** 3

**Excitement:**

3: Ambivalent: It has merits (e.g., it reports state-of-the-art results, the idea is nice), but there are key weaknesses (e.g., it describes incremental work), and it can significantly benefit from another round of revision. However, I won't object to accepting it if my co-reviewers champion it.

**Paper Topic And Main Contributions:**

The paper introduces a new dataset for researchers to improve upon in the field of multimodal learning. The benchmark includes eight datasets from various game genres, covering tabular, textual, and visual modalities, along with multiple baselines and experimental results from state-of-the-art classifiers. The paper also provides multi-aspect data analysis to demonstrate the diverse and multimodal-dependent properties of the dataset. This work might facilitate research on multimodal learning.

**Questions For The Authors:**

I don't quite understand why "Pidgey" is labeled as "Normal." When training a model on this sample, I'm uncertain whether the input-output mapping learned here can generalize to unseen data. For instance, can the model identify "Snorlax" as "Normal" based on the mapping learned from the Pidgey example?

**Reasons To Accept:**

The paper introduces a new benchmark dataset for multimodal learning, along with insights and experimental results that can guide future research in this area.

**Reasons To Reject:**

It is not clear to me how people can build a better multimodal model using the proposed new dataset. Perhaps it would be more convincing to demonstrate how benchmarks for real-world tasks improve when using a model trained on this dataset.

**Reproducibility:**

4: Could mostly reproduce the results, but there may be some variation because of sample variance or minor variations in their interpretation of the protocol or method.

**Reviewer Confidence:**

3: Pretty sure, but there's a chance I missed something. Although I have a good feel for this area in general, I did not carefully check the paper's details, e.g., the math, experimental design, or novelty.

---

> ### Author Rebuttal · Authors · 2023-08-27
>
> Dear Reviewer L86v,
>
> Thanks for reviewing our manuscript. We appreciate your time and comments. Our replies to the comments are as follows.
>
> ---
>
> ### Responses to "Reasons to Accept"
> > The paper introduces a new benchmark dataset for multimodal learning, along with insights and experimental results that can guide future research in this area.
>
> **Response**: We appreciate the positive comments.
>
> ---
>
> ### Responses to "Reasons to Reject"
> > It is not clear to me how people can build a better multimodal model using the proposed new dataset.
>
> **Response**: We believe that the creation of a new benchmark in an underrepresented domain, accompanied by easily interpretable quantitative evaluation metrics, serves as a foundational step in the development of robust and generalized multimodal models. Our MuG datasets stand out due to their unique composition of tabular-text-image modalities, and originating from the relatively unexplored video game domain.
> To address the concern, let us delve into the rationale behind our approach. Firstly, the scarcity of benchmarks encompassing all three of these modalities simultaneously is highlighted in both Section 1 and Section 2.1.  Our benchmark thus fills a significant hole for the multimodality learning community. Secondly, our focus extends beyond merely constructing a benchmark for refining game domain classifiers. It is about investigating strategies and techniques to effectively train models in scenarios where an abundance of pre-training samples is lacking. Overall, MuG facilitates rapid adaptation to novel data distributions, which is a critical aspect of real-world applicability.
> **Action**: We would add more discussion on the motivation and utility of MuG in the revised manuscript.
>
> > Perhaps it would be more convincing to demonstrate how benchmarks for real-world tasks improve when using a model trained on this dataset.
>
> **Response**: Thanks for the suggestion. It would be interesting to see if models pre-trained on MuG would deliver better performance on other datasets. However, we believe it is a bit unfair to say datasets and tasks for video games are not "real-world", so themselves are not worth studying. The whole video game industry is a huge real-world industry, and this work is filling a big hole for this industry, as thus for the multimodality learning community as well.
> Specifically, we would like to emphasize that this paper focuses on data collection and benchmarking. We manually curate and clean the datasets, and provide comprehensive data analysis to show how all modalities are necessary to solve the classification task. Regarding the benchmarking part, we evaluate six unimodal classifiers, two multimodal classifiers, and our proposed GNN-based multimodal classifier. Some subtasks remain challenging for powerful transformer-based models, which can be used to measure technical designs of multimodal classifiers. We believe even without considering how much algorithms developed for the video game industry can apply to other domains, our current contributions are significant regarding creating the first domain-specific benchmark in the particular industry.
>
> **Action**: We will add the idea of "pre-training on MuG, test on other benchmarks" as an important future direction in Sec.6 Conclusion.
>
> ---
>
> ### Responses to "Questions For The Authors"
> > I don't quite understand why "Pidgey" is labeled as "Normal."
>
> **Response**: Thank you for pointing out the confusion introduced by our toy example. Table 3 of Appendix B shows that each sample in the Pok\'emon dataset can have one or two types. For the specific sample *Pidgey* in Figure 1, its type_1 is *Normal*, and its type_2 is *Flying*. It is easy to induce Pedgey's type_2 from its bird-like appearance and abilities such as *Keen Eye*. The type_1 *Normal* is a bit tricky. In the world view of Pok\'emon, the *Normal* type is considered a *non-elemental* type. Since *Pidegey* has no abilities related to *Fire* or *Ice*, and it has no elemental-related appearance such as a burning fire on the tail, *Pidegey* is considered as *Normal* for its type_2.
> **Action**: We will modify the toy example in the revised manuscript. For *Pidgey*, we will use "type_2=*Flying*" as the illustration to make the toy example more accessible.
>
>
> > When training a model on this sample, I'm uncertain whether the input-output mapping learned here can generalize to unseen data. For instance, can the model identify "Snorlax" as "Normal" based on the mapping learned from the Pidgey example?''
>
> **Response**: When models infer whether *Snorlax* has a *Normal* type, models learn the shared patterns of training examples that are in the same type (*e.g.* no elemental related abilities or appearances).
> Moreover, experimental results support that multimodal classifiers can learn from the multimodal patterns of training samples for identifying Pok\'emon types. Table 2 (b) shows best classifier achieves *77.4* and *66.9* accuracy in type_1 and type_2 tasks, while the worst classifier only achieves *12.0* and *40.6*. The performance gap verifies that a better understanding of the multimodal input features delivers better performance.

---

### Official Review · Reviewer_SX92 · 2023-08-06

**Soundness:** 4

**Ethical Concerns:**

Yes

**Excitement:**

3: Ambivalent: It has merits (e.g., it reports state-of-the-art results, the idea is nice), but there are key weaknesses (e.g., it describes incremental work), and it can significantly benefit from another round of revision. However, I won't object to accepting it if my co-reviewers champion it.

**Justification For Ethical Concerns:**

- The authors mention that one of the sub-tasks is predicting the champion's skin category in the LoL game (line 175 and Table 3 and appendix). They do not provide additional context about what this target variable means in the context of the game (with which I am personally unfamiliar). It would be good to assess if this poses any ethical concerns. More broadly, there is a lack of discussion around other target variables within this benchmark dataset which makes it a bit opaque to evaluate (in terms of ethical concerns as well as usability for the community)
- The authors discuss the licensing of the data they have curated in B.1 (Appendix), but they do not mention the specific licenses for LoL and CS:GO.

**Paper Topic And Main Contributions:**

The authors propose a new dataset to aid the training and evaluation of multimodal models. The dataset comprises tables, text, and images from games. Additionally, the authors propose a new multimodal classification approach that models sample similarity in multimodal data using graph neural networks (GNNs). The authors conduct several analyses to provide insights into the datasets and do an exhaustive evaluation to compare the proposed approach with competitive baselines. The results demonstrate the effectiveness of their approach.

**Questions For The Authors:**

I would like to thank the authors for their work. I have the following questions that also elaborate on my concerns above.

A. The dataset and tasks were not clear to me. For instance, the target classes for each game made little to no sense to me, and I believe this will also be true for readers who are not familiar with the games. This is important because for the dataset to be adopted widely by the multimodal learning community, its structure should be interpretable to a wider audience. I would encourage the authors to include additional details about the task (for instance, what are the target classes (not simply stating their names but what it means -- for instance, predicting the "skin category/quality" could mean so many things without proper elaboration), and including qualitative examples for each of the eight tasks).

B. Related to my previous comment, it was not clear to me how important is evaluation on datasets from games. I could imagine a similar tabular+text+image dataset being curated from Wikipedia (WIT does this to some extent: https://github.com/google-research-datasets/wit). It would help if the authors discussed the motivation for a game-based dataset and the challenges/opportunities it presents that datasets from other domains cannot present.

C. Besides a new dataset, the authors propose a new approach to MugNET. An ideal evaluation of the effectiveness of the proposed approach would involve comparing its performance against baselines on existing benchmark datasets. However, the authors limit their evaluation to the new dataset. This limits the inferences about the applicability of the proposed approach as it is not empirically shown to generalize to other tasks. I would encourage the authors to specifically motivate why the tasks within the proposed benchmark warrant a new approach and how MugNET delivers on those requirements.

D. During data cleaning, the authors mention that they group all the classes with small representation in the overall dataset into one single target class. I wonder how does this compare to dropping these classes? In the former approach, the grouped target class could have high heterogeneity, which could raise questions around the purpose of that class.

**Reasons To Accept:**

- The new dataset is publicly available and will add a new dimension to multimodal research (including tables beyond the usual vision and language modalities). The authors also claim that the dataset will be publicly available.
- The authors also propose a new multimodal learning approach that models sample-similarity using within-modality graph neural networks and then aggregates the modality-specific representations.
- The authors show experiments with several baselines and demonstrate the competitiveness of their proposed approach on the new dataset.

**Reasons To Reject:**

- My major concern with the paper is that the dataset is not well-motivated, and several crucial details that are needed to understand the datasets and tasks are missing from the paper
- The proposed approach is only evaluated on the new dataset, and it is not clear how effective it will be on existing tasks over the baselines considered in the paper
- Some of the design choices made while curating the dataset are not well-motivated/discussed

**Reproducibility:**

4: Could mostly reproduce the results, but there may be some variation because of sample variance or minor variations in their interpretation of the protocol or method.

**Reviewer Confidence:**

4: Quite sure. I tried to check the important points carefully. It's unlikely, though conceivable, that I missed something that should affect my ratings.

**Typos Grammar Style And Presentation Improvements:**

- The anonymous link to the benchmark (in the abstract) was not accessible to me
- Line 189 seems to have a typo: diversity instead of diverse?
- Figure 4 overlaps with some of the text above it
- A broader comment about the structure of the paper: several critical details about the dataset and proposed approach are in the appendix instead of in the main paper. I would suggest the authors move some of the analyses to the appendix to be able to keep the technical details of the dataset and the approach in the main paper, as those are the key contributions of this work

---

> ### Author Rebuttal · Authors · 2023-08-28
>
> Dear Reviewer SX92,
>
> Thanks for your careful consideration. We greatly appreciate the positive comments and address the concerns below.
>
> ---
>
> ### Responses to "Reasons to Accept"
>
> **Response**: Thank you for acknowledging our contributions on the benchmark creation, data analysis, proposed baselines, and comprehensive experiments.
>
> ---
>
> ### Responses to "Reasons to Reject"
>
> We will address each point along with its related questions in "Questions For The Authors".
>
> > My major concern with the paper is that the dataset is not well-motivated.
>
> > (Q.B) Related to my previous comment, it was not clear to me how important is evaluation on datasets from games. I could imagine a similar tabular+text+image dataset being curated from Wikipedia (WIT does this to some extent: https://github.com/google-research-datasets/wit). It would help if the authors discussed the motivation for a game-based dataset and the challenges/opportunities it presents that datasets from other domains cannot present.
>
> **Response**: We appreciate your thoughtful feedback on the motivation behind the MuG benchmark.
> MuG's significance extends beyond merely constructing a benchmark for enhancing game domain classifiers. It is also beneficial to the investigation of innovative strategies and techniques in underrepresented domains where the large-scale pre-training samples are limited, and where novel data distributions are prevalent.
> Importantly, as highlighted in Sections 1 and 2.1, MuG addresses a critical gap in the realm of tabular-text-image multimodal classification. Existing benchmarks rarely encompass all three modalities simultaneously, and this gap is crucial in reflecting the diversity of real-world applications. For instance, tasks like 48-hour mortality prediction in HAIM-MIMIC-MM or adoption prediction in PetFinder, showcase the practical relevance of MuG.
> Furthermore, our decision to collect data from the video game domain is grounded in the objective of introducing diversity to the multimodal learning community. This domain offers unique challenges and opportunities that datasets from other general domains, such as Wikipedia, may not provide.
>
> **Action**: We will certainly enhance the discussion in the Introduction and Related Work sections to elaborate on these motivations and distinctions.
>
> > several crucial details that are needed to understand the datasets and tasks are missing from the paper.
>
> > (Q.A.) The dataset and tasks were not clear to me. For instance, the target classes for each game made little to no sense to me, and I believe this will also be true for readers who are not familiar with the games. This is important because for the dataset to be adopted widely by the multimodal learning community, its structure should be interpretable to a wider audience. I would encourage the authors to include additional details about the task (for instance, what are the target classes (not simply stating their names but what it means -- for instance, predicting the "skin category/quality" could mean so many things without proper elaboration), and including qualitative examples for each of the eight tasks).
>
> **Response**: Thank you for the valuable suggestion. Please note again that one major motivation for us to collect data from video games for prediction tasks there is to bring diversity to the multimodal learning community. That being said, we should have better elaborated the prediction targets and provided more real instances, to make MuG more accessible for potential readers and researchers who are not familiar with the incorporated video games.
>
> **Action**: Regarding the prediction targets, we plan to add the following detailed elaboration about the source video games and the prediction targets in Appendix B.1.
>
> - Pok\'emon: Pok\'emon is a popular video game centered around fictional creatures called "Pocket Monsters" that trainers capture and train to battle each other. Regarding the prediction targets, Pok\'emons are categorized into various elemental **types**, such as Fire, Ice, Normal (non-elemental), and more. Each Pok\'emon can have one primary type (for pkm_t1) and one secondary type (for pkm_t2).
> - HearthStone: HearthStone is an online collectible card game developed by Blizzard Entertainment, featuring strategic gameplay where players build decks and compete against each other using a variety of spells, minions, and abilities. Regarding the prediction targets, each card belongs to one certain **category** (for hs_ac) such as minon, spell, weapon, etc. Each card is part of a **set** (for hs_as) where new card sets are released periodically to introduce new content and strategies to the game. Each minion card is essentially a creature, thus it can be divided into different **races** (for hs_mr). Each spell card. Similarly to the minion race, each spell card belongs to a specific **school** (for hs_ss) such as Shadow, Nature, *etc*.
> - League of Legends (LoL): League of Legends is a widely popular multiplayer online battle arena (MOBA) video game developed by Riot Games where teams of players compete in fast-paced matches, utilizing unique champions with distinct abilities to achieve victory. Regarding the prediction targets, a champion skin in LoL is a cosmetic alteration to the appearance of the champion. Depending on the rarity and price, a champion's skin belongs to a specific **category** (for lol_sc).
> - "Counter-Strike: Global Offensive": CS:GO is a popular multiplayer first-person shooter video game developed by Valve and Hidden Path Entertainment, where players join teams to compete in objective-based matches involving tactical gameplay and precise shooting. Similarly, skins alter the appearance of weapons. The prediction target is the **skin quality** according to its rarity (for csg_sq).
>
> Furthermore, we will provide more real instances of the eight tasks in MuG.
>
> > The proposed approach is only evaluated on the new dataset, and it is not clear how effective it will be on existing tasks over the baselines considered in the paper
>
> > (Q.C) Besides a new dataset, the authors propose a new approach to MugNET. An ideal evaluation of the effectiveness of the proposed approach would involve comparing its performance against baselines on existing benchmark datasets. However, the authors limit their evaluation to the new dataset. This limits the inferences about the applicability of the proposed approach as it is not empirically shown to generalize to other tasks. I would encourage the authors to specifically motivate why the tasks within the proposed benchmark warrant a new approach and how MugNET delivers on those requirements.
>
> **Response**: It's important to clarify that our submission falls under the "Resources and Evaluation" track, where the primary focus is on the creation and benchmarking of the MuG dataset. MuGNet is introduced as one of the baseline methods, not really an important novel design. That being said, we understand your concern regarding the limited evaluation scope on MuGNet. Our intention is to avoid diverting attention from the core contribution of this paper. We agree that a thorough evaluation of MuGNet's performance on existing benchmark datasets is an important aspect that merits further investigation, such as in a follow-up research paper.
>
> > Some of the design choices made while curating the dataset are not well-motivated/discussed
>
> > D. During data cleaning, the authors mention that they group all the classes with small representation in the overall dataset into one single target class. I wonder how does this compare to dropping these classes? In the former approach, the grouped target class could have high heterogeneity, which could raise questions around the purpose of that class.
>
> **Response**: The suggested dropping strategy is an interesting alternative for data cleaning. We did not try that during our data cleaning. Indeed, we only conduct grouping strategy on *hs_ac* and *lol_sc*. The number of samples in sparse classes is relatively small in MuG.
>
> ---
>
> ### Response to "Typos Grammar Style And Presentation Improvements"
>
> > The anonymous link to the benchmark (in the abstract) was not accessible to me
>
> **Action**: We have tested the anonymous link and it is accessible now (sometimes the anonymous link may be out of work due to the downtime of the anonymous Github service).
>
> > Line 189 seems to have a typo: diversity instead of diverse?
>
> **Action**: Thanks for pointing out the typo. We will fix it and carefully check for other typos in the revised version.
>
> > Figure 4 overlaps with some of the text above it
>
> **Action**: We will fix the overlapping issue in the revised manuscript.
>
> > A broader comment about the structure of the paper: several critical details about the dataset and proposed approach are in the appendix instead of in the main paper. I would suggest the authors move some of the analyses to the appendix to be able to keep the technical details of the dataset and the approach in the main paper, as those are the key contributions of this work
>
> **Action**: Thanks for the valuable suggestion. We will be able to add more technical details to the main paper when submitting the camera-ready version.
>
> ---
>
> ### Response to "Justification For Ethical Concerns"
>
> > The authors mention that one of the sub-tasks is predicting the champion's skin category in the LoL game (line 175 and Table 3 and appendix). They do not provide additional context about what this target variable means in the context of the game (with which I am personally unfamiliar). It would be good to assess if this poses any ethical concerns. More broadly, there is a lack of discussion around other target variables within this benchmark dataset which makes it a bit opaque to evaluate (in terms of ethical concerns as well as usability for the community)
>
> **Response**: Thank you for bringing this up. We have provided a detailed elaboration of the prediction targets in "Resposne to (Q.B)". The champion skins in LOL are stylish decorations that have nothing to do with races/genders or other unethical variables.
>
> > The authors discuss the licensing of the data they have curated in B.1 (Appendix), but they do not mention the specific licenses for LoL and CS:GO.
>
> **Response**: For the data source of LoL and CS:GO, all data are collected from public web content, and no user-specific private information is collected.

---

### Official Review · Reviewer_TBbD · 2023-08-10

**Soundness:** 4

**Excitement:**

4: Strong: This paper deepens the understanding of some phenomenon or lowers the barriers to an existing research direction.

**Missing References:**

In general, I believe there should be more discussion about what MUG provides that the listed existing datasets can't provide in section 2.1. Also, while the authors included several domain-specific multimodal benchmarks, there are also existing benchmarks for multimodal classifications that are more general and not restricted to certain modality combinations that needs to be compared to MUG, such as MultiBench [1] and MMBench[2].

[1] https://arxiv.org/pdf/2107.07502.pdf
[2] https://arxiv.org/pdf/2212.01241.pdf

**Paper Topic And Main Contributions:**

This paper presents a new benchmark for multimodal classification benchmark based on game data. The data is collected from 4 vastly different video games with 8 total classification tasks, and the tasks involves information from 3 modalities (tabular, text and visual). There are features that are naturally missing for some data points, which made the tasks more challenging. The authors then benchmarked several different unimodal and multimodal models on the tasks, including several well-known models as well as a new model called MugNet. The experiments showed that (1) the tasks do require all modalities to solve well and (2) while existing models can do reach a decent performance, some tasks remains challenging.

**Questions For The Authors:**

A. It's a bit uncommon to use a graph-based neural network for classification tasks on tabular/visual/text tasks. What is the reason/intuition for using a graph-based architecture for MUGNet?

**Reasons To Accept:**

This paper presents a valuable new dataset for multimodal classification problems. The dataset contains data from 3 different modalities, and some features are naturally missing which makes the tasks even more challenging. It is also uncommon to have a multimodal classification dataset from the video games domain, which makes this dataset more valuable for the research community.

The authors included comprehensive analysis on the data and features of the dataset to show that all modalities are needed to solve the tasks. Moreover, they benchmarked the dataset on several well-known unimodal and multimodal models, and demonstrated how the dataset/benchmark can be used to comprehensively evaluate multimodal classification models, including performance and performance-speed tradeoffs. Also, on several tasks, even the best-performing models are still quite far from perfect, meaning that the tasks remains challenging for future researchers to explore.

**Reasons To Reject:**

In the experiments, some of the tasks (such as lol_sc) can already be almost perfectly solved by benchmarked models, which makes them less challenging or valuable for the research community (although most of the tasks still remain challenging).



**Reproducibility:**

3: Could reproduce the results with some difficulty. The settings of parameters are underspecified or subjectively determined; the training/evaluation data are not widely available.

**Reviewer Confidence:**

3: Pretty sure, but there's a chance I missed something. Although I have a good feel for this area in general, I did not carefully check the paper's details, e.g., the math, experimental design, or novelty.

**Typos Grammar Style And Presentation Improvements:**

Figure 4 is overlapped with the last row of text above it (line 324). Also, Figure 4 is too small for readers to see the content details.

---

> ### Author Rebuttal · Authors · 2023-08-28
>
> Dear Reviewer TBbD,
>
> We'd like to thank you for your careful review and valuable comments. We believe the constructive feedback will improve the manuscript and increase its potential impact on the community.
>
> ---
>
> ### Responses to "Reasons to Accept"
>
> > It is also uncommon to have a multimodal classification dataset from the video games domain, which makes this dataset more valuable for the research community.
> > The authors included comprehensive analysis on the data and features of the dataset to show that all modalities are needed to solve the tasks.
>
> **Response**: We thank the reviewer for the encouraging comments.
>
> ---
>
> ### Responses to "Reasons to Reject"
>
> > In the experiments, some of the tasks (such as lol_sc) can already be almost perfectly solved by benchmarked models, which makes them less challenging or valuable for the research community (although most of the tasks still remain challenging).
>
> **Response**: Thanks for pointing out that the *loc_sc* task appears relatively easy for existing multimodal classifiers, with the best benchmarked model achieving an accuracy of 96.8%. Although there is still room to improve, we agree that this particular task is relatively easy. Despite this, the remaining five tasks in the dataset still pose significant challenges. Your insight will guide us in designing future benchmarks that strike the right balance between being challenging and valuable for advancing the field. Thank you for your constructive input.
>
> ---
>
> ### Responses to "Questions For The Authors"
>
> > It's a bit uncommon to use a graph-based neural network for classification tasks on tabular/visual/text tasks. What is the reason/intuition for using a graph-based architecture for MUGNet?
>
> **Response**: The decision to opt for a graph-based architecture stems from the exciting potential demonstrated by Graph Neural Networks (GNNs) in various classification tasks. While GNNs have been predominantly applied in graph-related settings, inspired by the rationale that samples with similar tabular/textual/visual features tend to be assigned with the same category label, we believe that MuGNet is powerful and efficient in capturing such signals and it provides a novel perspective to solving multimodal classification tasks. Our empirical results also support this intuition.
>
> **Action**: In the revised manuscript, we will add discussions as such to motivate the design of MuGNet in Sec.4.2.
>
> ---
>
> ### Responses to "Missing References"
>
> > In general, I believe there should be more discussion about what MUG provides that the listed existing datasets can't provide in section 2.1. Also, while the authors included several domain-specific multimodal benchmarks, there are also existing benchmarks for multimodal classifications that are more general and not restricted to certain modality combinations that needs to be compared to MUG, such as MultiBench [1] and MMBench[2].
>
> **Response**: We greatly appreciate your input and the references you provided. Both MultiBench and MMBench are relevant works. MultiBench is indeed an insightful work, offering a diverse range of 15 multimodal datasets. However, it's important to note that MultiBench lacks tasks that encompass the combination of tabular, textual, and image modalities simultaneously. Regarding MMBench, we didn't include it due to its release date in August 2023, which fell after the EMNLP submission deadline.
> As Sec.2.1 states, there exist very few tabular-text-image classification benchmarks. Among them, PetFinder focuses on pets, and HAIM-MIMIC-MM focuses on the healthcare domain. MugNet provides another rare video game domain. This serves as an essential contribution to diversifying the available benchmarks and addressing underexplored areas.
>
> **Action**: We will cite and discuss the missing references in the revised manuscript.
>
> ---
>
> ### Response to "Typos Grammar Style And Presentation Improvements"
>
> > Figure 4 is overlapped with the last row of text above it (line 324). Also, Figure 4 is too small for readers to see the content details.
>
> **Action**: We will fix the overlapping issue of Figure 4 in the revised manuscript. Regarding the size issue, we will be able to provide a larger version of Figure 4 in the revised manuscript.

---

### Meta-Review · Area_Chair_NDWC · 2023-09-16

**Recommendation:** 4

**Metareview:**

**Summary:**
This paper presents a new benchmark in the field of multimodal learning for multimodal classification, leveraging on game data. The data is collected from four vastly different video games with a total of eight classification tasks, each tasks involves information from three modalities: tabular, text and visual data.
The authors conduct benchmarks involving different unimodal and multimodal models, including several well-known ones. Additionally, the authors propose a new multimodal classification approach, called MugNet, which models sample similarity in multimodal data using graph neural networks (GNNs).
The authors execute an exhaustive evaluation to compare the proposed approach against competitive baselines. The results demonstrate the effectiveness of their approach. This work might facilitate research on multimodal learning.

**Strengths:**
The reviewers acknowledge the following strengths in this contribution:
1. The paper introduces a valuable new dataset for addressing multimodal classification challenges, with the authors committed to making the dataset publicly available.
2. The rarity of a multimodal classification dataset derived from the domain of video games enhances its significance within the research community.
3. The authors propose an innovative multimodal learning approach that leverages within-modality graph neural networks to model sample similarity and subsequently aggregates modality-specific representations.
4. Comprehensive data and feature analyses are included in the paper, illustrating the indispensability of all modalities for solving the tasks.
5. The authors conduct benchmarking on the dataset using various well-established unimodal and multimodal models, highlighting how the dataset and benchmark can facilitate comprehensive evaluations of multimodal classification models, encompassing performance and performance-speed tradeoffs.
6. The experimental results offer valuable insights for guiding future research in this domain.

**Weaknesses:**
Reviewers share certain concerns with this paper. The dataset lacks clear motivation, and critical details necessary for comprehending the datasets and tasks are absent. The proposed approach is solely evaluated on the new dataset, and its effectiveness against existing tasks and baseline models considered in the paper remains unclear. It might be more persuasive to showcase the improvement in benchmarks for real-world tasks achieved by employing a model trained on this dataset. Additionally, some of the tasks in the experiments can already be solved nearly perfectly by the benchmarked models

**Author-Reviewer discussion and acknowledgment:**
The authors provided clarifications addressing the concerns raised by the reviewers and have delineated the improvements to be made, during the rebuttal response and discussion phase. All reviewers have responded and acknowledged the authors' arguments.

**Conclusion:**
The paper is well-structured, but reviewers suggest that the authors rectify the identified typos. Furthermore, reviewers recommend that the authors include additional references.

---

### Decision · Program_Chairs · 2023-10-07

**Decision:**

Accept-Findings

**Comment:**

**Summary:**
This paper presents a new benchmark in the field of multimodal learning for multimodal classification, leveraging on game data. The data is collected from four vastly different video games with a total of eight classification tasks, each tasks involves information from three modalities: tabular, text and visual data.
The authors conduct benchmarks involving different unimodal and multimodal models, including several well-known ones. Additionally, the authors propose a new multimodal classification approach, called MugNet, which models sample similarity in multimodal data using graph neural networks (GNNs).
The authors execute an exhaustive evaluation to compare the proposed approach against competitive baselines. The results demonstrate the effectiveness of their approach. This work might facilitate research on multimodal learning.

**Strengths:**
The reviewers acknowledge the following strengths in this contribution:
1. The paper introduces a valuable new dataset for addressing multimodal classification challenges, with the authors committed to making the dataset publicly available.
2. The rarity of a multimodal classification dataset derived from the domain of video games enhances its significance within the research community.
3. The authors propose an innovative multimodal learning approach that leverages within-modality graph neural networks to model sample similarity and subsequently aggregates modality-specific representations.
4. Comprehensive data and feature analyses are included in the paper, illustrating the indispensability of all modalities for solving the tasks.
5. The authors conduct benchmarking on the dataset using various well-established unimodal and multimodal models, highlighting how the dataset and benchmark can facilitate comprehensive evaluations of multimodal classification models, encompassing performance and performance-speed tradeoffs.
6. The experimental results offer valuable insights for guiding future research in this domain.

**Weaknesses:**
Reviewers share certain concerns with this paper. The dataset lacks clear motivation, and critical details necessary for comprehending the datasets and tasks are absent. The proposed approach is solely evaluated on the new dataset, and its effectiveness against existing tasks and baseline models considered in the paper remains unclear. It might be more persuasive to showcase the improvement in benchmarks for real-world tasks achieved by employing a model trained on this dataset. Additionally, some of the tasks in the experiments can already be solved nearly perfectly by the benchmarked models

**Author-Reviewer discussion and acknowledgment:**
The authors provided clarifications addressing the concerns raised by the reviewers and have delineated the improvements to be made, during the rebuttal response and discussion phase. All reviewers have responded and acknowledged the authors' arguments.

**Conclusion:**
The paper is well-structured, but reviewers suggest that the authors rectify the identified typos. Furthermore, reviewers recommend that the authors include additional references.